# Influence of Polymer Types on the Mechanical Properties of Polymer-Modified Cement Mortars

**Min Ook Kim**

Coastal Development and Ocean Energy Research Center, Korea Institute of Ocean Science and Technology, Busan 49111, Korea; minookkim@kiost.ac.kr; Tel.: +82-51-664-3520

**Abstract:** This is an experimental study showing the effect of four types of polymers (acrylate polymer (AC), polyvinyl alcohol (PVA), styrene–butadiene rubber (SBR), and ethylene-vinyl acetate (EVA)) on the mechanical properties of polymer-modified cement mortars (PCMs). One polymer dosage was used in this study (3%), and the effect of this dosage on PCMs was compared with a control mortar mix with 0% polymer. The compressive, flexural, and pull-off bond strengths were measured and compared with previous results in works of literature. The effect of polymer addition on improving the mechanical properties of PCMs was clarified, and this effect was more obvious on the flexural strength than that on the compressive strength. The PCMs containing EVA showed the best performance, with up to 33% and 63% increases in compressive and flexural strengths after 28 days, respectively. In comparison, AC, PVA, and SBR produced smaller (16%–46% compared to control) improvements in the flexural strength after 28 days. In general, PCMs containing EVA showed the best mechanical properties.

**Keywords:** cement mortar; polymers; compressive strength; flexural strength; pull-off bond strength; failure modes

## 1. Introduction

It is known that adding a polymeric compound to a cementitious material can improve the material properties such as strength, durability, and interfacial bond between the newly applied material and the existing substrate [1–12]. Therefore, incorporating polymers in patch repair cement mortars is common for better repair work of various reinforced concrete (RC) structures, including bridges, dams, high-rise buildings, pavements, and marine concrete structures [13–21]. There exist a few representative types of polymers, including acrylate polymers (AC), polyvinyl alcohol (PVA), styrene–butadiene rubber (SBR), and ethylene-vinyl acetate (EVA), all of which are utilized in the repair or surface treatment of RC structures. Many experimental studies were conducted to investigate the effect of different polymer types on the mechanical properties and durability of polymer-modified cement mortars (PCMs) [1–9]. Doğan and Bideci [1] studied the effects of different SBR replacement ratios (by weight) on the concrete strength measured at three, seven, and 28 days. Furthermore, they reported that the 1% replacement showed a higher strength compared to the 3%, 5%, and 8% replacements. Doğan and Bideci [1] also confirmed a significant reduction in the measured strength of the sample with 8% SBR replacement. These results are consistent with those reported by Shafieyzadeh [2]. Shafieyzadeh [2] reported that the SBR replacement ratio should be less than 5% to improve the mechanical properties of PCM. Ariffin et al. [3] stated that the strength of the epoxy-modified cement mortar could be reduced when the epoxy replacement ratio was higher than 10%, based on an experimental study focusing on the strength properties of cement mortars. Maranhão and John [4] investigated the effect of environmental conditions on the adhesive strength and deformation of various PCMs and confirmed that the constant laboratory conditions were preferred to outside exposure when demonstrating the effects of test

variables. Li et al. [5] recently conducted an experimental study to clarify the effects of three different polymers (SBR, styrene–acrylic ester (SAE), and polyacrylic ester (PAE)) on the mechanical properties and durability performance of PCMs, and they concluded that PCMs with SBR exhibited better performance when compared to PCMs with SAE or PAE, in terms of strength, weight loss, and chloride ion penetration resistance. Aggarwal et al. [6] compared two PCMs containing epoxy emulsion and acrylic emulsion, and they reported that the PCM using epoxy emulsion showed better performance in terms of strength and durability. Mirza et al. [7] conducted an extensive experimental study to compare two different polymers (SBR and acrylics) through measurements of strength, durability, and bond strength. The authors concluded that the bond strength of PCMs containing SBR was generally higher than that of PCM containing acrylics. By contrast, Al-Zahrani et al. [8] stated that no significant differences were observed between PCMs and conventional cement mortar in terms of mechanical properties and penetration resistance, based on test results and comparisons. Schulze [9] reported that water-to-cement ratios (0.38 to 0.47) did not show a significant influence on the adhesion between the PCM and substrate concrete.

A few case studies also confirmed that the application of PCMs to structural members can bring a positive effect to the structural performance of repaired RC structures [13–15]. Ahmad et al. [13] studied the possibility of using PCMs containing an organic film-forming re-dispersible polymeric powder produced in Pakistan for repairing cracks that occurred on the surface of RC beams. The authors reported that the repaired specimens showed an equivalent or higher load-carrying capacity under the same load condition. Chen and Won [14] reported the results of a long-term investigation on patch-repaired concrete pavement structures with a focus on the interfacial bond between the PCM and the existing substrate. Specifically, the authors investigated three commercial products (FastPatch DPR (Willamette Valley Company, Eugene, OR, USA), SILSPEC FLEXPATCH (Silicone Specialties Inc., Tulsa, OK, USA), and Sikadur-72 JNS (Sika, Lyndhurst, NJ, USA)) in the study. Chen and Won [14] concluded that the use of both polymer and fiber was effective for the longer service life of structures while reducing spalling and cracks. Kim et al. [15] confirmed that the high bond strength of applied PCMs could contribute to improved cracking resistance by sustaining a higher maximum load. Some researchers focused on the long-term mechanical properties of PCMs [22–24]. Fischer et al. [22] recently investigated the long-term creep behavior of two polymers (epoxy and vinyl ester (VE)), and they reported that the moisture content for VE resin can influence the creep modulus. Berardi and Mancusi [23] developed a mechanical model to predict the long-term behavior of reinforced polymer concrete beams. Berardi and Mancusi [24] also studied the time-dependent changes in structural behavior of reinforced polymer concrete columns, and they reported that the viscosity of polymers may affect the integrity and serviceability of structural members. However, difficulties still exist in the selection of an appropriate polymer, given that too many products are available commercially while their performance is not fully understood. Therefore, understanding the influence of polymer type on the mechanical properties of PCMs is important for the selection of the polymer and improved repair work of old and deteriorated RC structures. In this regard, this study aims to provide some useful information for the selection of appropriate polymer types throughout the experimental campaign. Four different types of polymers with two different dosages (0%, 3%) were employed to prepare samples, and the effects of the polymer types on the mechanical properties of PCMs were studied. The addition rate of 3% was determined based on the results in works of literature [1,2].

## 2. Research Significance

Although various types of polymeric admixtures were developed and used in repair work for existing RC structures, the mechanical properties of PCMs are not well understood. In addition, the experimental studies comparing commercial polymeric products that can be applied to improve the mechanical properties of cement mortar are still lacking. It is expected that the performance of PCMs can be varied depending on the manufacturers; thus, civil engineers need to select the optimum

material commercially available, such that the use of polymeric admixture can ultimately be linked to the long-term performance of RC structures. Understanding the mechanical properties of various PCMs can then allow engineers to grasp the important aspects of determining the optimum polymer type and replacement ratio. To this end, an experimental set-up was arranged, and a comprehensive experimental study was carried out to investigate the effects of different polymer types and cement replacement ratios on the mechanical properties of PCMs. The compressive, flexural, and pull-off tensile bond strength was measured, and critical comparisons with previous studies were made.

## 3. Materials and Methods

The effects of four different polymer types and two different replacement rations on the performance of PCMs were considered for this study. The measured values include compressive, flexural, and pull-off bond strength. The detailed test materials, variables, measurements, and procedures are described in the sections below.

### 3.1. Materials and Test Variables

Four different commercially available polymers were investigated in this study. Tables 1–3 show the chemical and mineral compositions of Type I ordinary Portland cement (OPC) satisfying the Korean Standard (KS), the particle size distribution of fine aggregate, and properties of polymers, respectively. The mixture proportions of the PCMs and substrate concrete are summarized in Table 4. The PCM specimens were cast with a cement–fine aggregate ratio of 0.5, a water–cement ratio of 0.4, and a polymer–cement ratio of 0 or 0.03. The amount of water was determined based on previous studies [14].

**Table 1.** Chemical and mineral compositions of ordinary Portland cement.

| Chemical (% by weight) | | | | | | | Mineral (% by weight) | | |
|---|---|---|---|---|---|---|---|---|---|
| $Al_2O_3$ | CaO | $SO_3$ | $SiO_2$ | $Fe_2O_3$ | MgO | Loss of ignition | $C_3S$ | $C_2S$ | $C_3A$ |
| 5.0 | 62.3 | 2.1 | 20.5 | 3.4 | 3.6 | 2.4 | 53.1 | 18.5 | 7.8 |

**Table 2.** Sieve analysis of silica sand used.

| Sieve Size | Mass Retained (%) | Cumulative Mass Retained (%) |
|---|---|---|
| 1.0 mm | 0.00 | 0.00 |
| 850 µm | 0.01 | 0.01 |
| 600 µm | 1.20 | 1.21 |
| 425 µm | 6.50 | 7.71 |
| 300 µm | 28.00 | 35.71 |
| 212 µm | 33.94 | 69.65 |
| 150 µm | 21.05 | 90.70 |
| 106 µm | 7.50 | 98.20 |
| 75 µm | 1.50 | 99.70 |
| Pan | 0.30 | 100.00 |

The substrate concrete specimens were prepared with a cement content of 340 kg/m$^3$ and a water–cement ratio of 0.24 to produce a high strength based on the result of previous research [20]. The averaged compressive strength of the substrate concrete measured at 28 days was 82.3 MPa. As illustrated in Table 4, four different polymers, namely, acrylic, PVA, SBR, and EVA, with two different cement replacement ratios (0%, 3%), were added to modify cement mortars.

**Table 3.** Properties of selected polymers (N/A—not available; AC—acrylate polymer; PVA—polyvinyl alcohol; SBR—styrene–butadiene rubber; EVA—ethylene-vinyl acetate).

|  | AC | PVA | SBR | EVA |
|---|---|---|---|---|
| Solid content (%) | 99 ± 1 | 99 ± 1 | 48 ± 1 | 99 ± 1 |
| Physical appearance | Powder | Powder | Latex | Powder |
| pH value | N/A | N/A | 9.9–10.5 | N/A |
| Viscosity (mPa·s) | N/A | N/A | 40 | N/A |
| Density (g/cm$^3$) | 0.46 | 0.5 ± 0.1 | 1.04 | 0.52 |
| Glass transition temperature (°C) | N/A | 14 | 14 | N/A |
| Minimum film-forming temperature (°C) | <5 | <0 | <4 | <4 |

**Table 4.** Mix proportions for polymer-modified cement mortars and substrate concrete.

| Mix Proportions of Polymer-Modified Cement Mortars (Mass Ratio) | | | | | |
|---|---|---|---|---|---|
| Water | Cement | Fine Aggregate | Water/Cement | Sand:Cement | Polymer Content (%) |
| 40 | 100 | 200 | 0.4 | 5:1 | 0 |
| 40 | 97 | 200 | 0.4 | 5:1 | 3 |
| Concrete Mix Proportion | | | | Compressive Strength | |
| Water (kg/m$^3$) | Cement (kg/m$^3$) | Fine Aggregate (kg/m$^3$) | Coarse Aggregate (kg/m$^3$) | Water Reducer (kg/m$^3$) | $f'_c$ (MPa) |
| 83 | 340 | 325 | 451 | 5.0 | 82.3 |

*3.2. PCM and Composite Sample Fabrication*

The PCM and composite sample fabrications were conducted according to the following procedure:

1. Rectangular concrete slabs with dimensions of 300 × 300 × 30 mm were cast and steam-cured under controlled room conditions.
2. Substrate concrete slabs were then placed in water for more than 24 h before applying the PCMs to create a saturated surface dry (SSD) condition.
3. The polymer was initially mixed with tap water, and a fixed amount of cement was added to produce the PCMs.
4. For the compressive and flexural strength tests, 50-mm cubic specimens and prismatic specimens with dimensions of 40 × 40 × 160 mm were cast, respectively, using an external vibrator.
5. For the pull-off bond test, the mixed PCM was applied to the top of the concrete slab, and surface treatment was carried out using a spatula trowel.
6. The prepared samples were cured at 20 °C and 60% relative humidity until the age of testing.

*3.3. Test Procedures*

Mechanical tests were carried out on each PCM, as well as on the substrate concrete. Figure 1a–c show the compressive, flexural, and pull-off bond test set-ups, respectively. The compressive strengths of the PCMs were determined from three standard 50-mm cubes as per ASTM C 109 [25]. The flexural strengths of the PCMs were measured using three prismatic specimens of 40 × 40 × 160 mm as per ASTM C 348 [26]. The specimens underwent flexural tests at a central point of loading as shown in Figure 1b. The applied loading rates for the compressive and flexural strength measurements were 900 N/s and 0.02 mm/min, respectively, and the tests were conducted at seven and 28 days after the date of casting. Before conducting the bond strength measurement test, the applied PCM layer was cut

into 40-mm square specimens, and dollies were attached using epoxy glue to the targeted locations, as shown in Figure 1d.

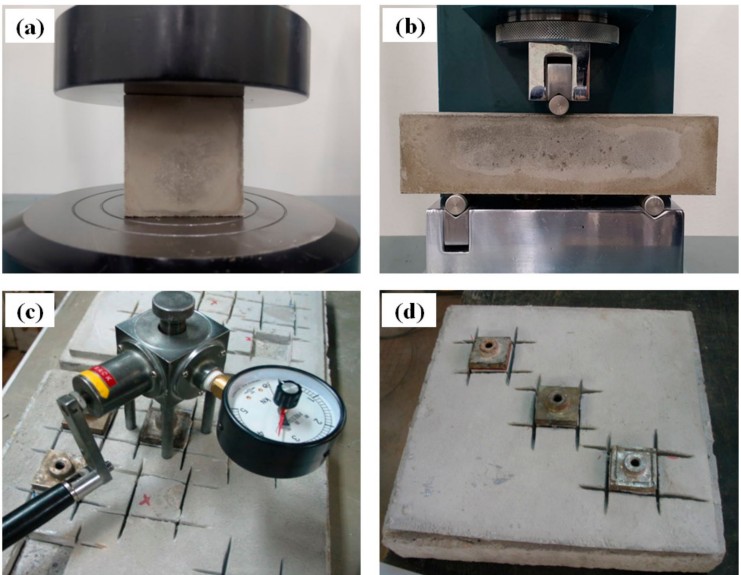

**Figure 1.** Mechanical test set-ups: (**a**) compressive test; (**b**) flexural test; (**c**) pull-off bond test; (**d**) attached dollies on the surface of the polymer-modified cement mortar (PCM) overlay.

The pull-off bond strengths of the PCMs were determined from five samples as per ASTM C 1583 [27]. The pull-off rate adopted in this study was 0.02 MPa/s based on previous research [20].

## 4. Test Results

### 4.1. Effect of Polymer Type on Compressive Strength

Table 5 and Figure 2 show the compressive strength results for the PCMs, and the result of the control specimen is included in the same figure for better comparison. PCM samples ranked EVA, PVA, SBR, and AC in compressive strength from high to low. The PCM containing EVA showed the highest compressive strength at 28 days and was approximately 33% higher than that of the control specimen. Furthermore, the estimated $p$-value between the control and the PCM containing EVA was less than 0.05. The compressive strength of the PCM containing PVA was 48.9 MPa, and this value was quite similar to the previous result of 48.7 MPa reported by Kim et al. [15]. The PCM containing SBR showed a 36% higher compressive strength than that of PCM containing AC, and this result shows good agreement with previous studies reported by Li et al. [5] and Medeiros et al. [28]. By contrast, the 28-day compressive strength of the PCM containing AC was 14% lower compared to that of the control specimen.

**Table 5.** Averaged compressive strengths measured at seven and 28 days.

| Compressive Strength (Days) | Control | AC | PVA | SBR | EVA |
|:---:|:---:|:---:|:---:|:---:|:---:|
| | | | MPa | | |
| 7 | 35.15 | 27.32 | 41.61 | 34.86 | 44.03 |
| 28 | 37.72 | 32.34 | 49.91 | 44.14 | 50.27 |

All PCM samples exhibited a rapid increase in compressive strength between seven and 28 days when compared to that of the control specimen. Specifically, the increase in strength of the PCM samples between seven and 28 days ranged from 15% to 25%, while the control specimen showed only a 5% increase in measured strength. This difference might be attributed to the effects of the incorporated

polymers. It can be seen that the PCM containing SBR showed a slower strength development compared to others. The reason for the low compressive strengths from the PCM containing AC at seven and 28 days is not clear at this time. However, some researchers pointed out that the inclusion of polymer in cement mortar can reduce the strength due to a lower mechanical capacity of polymer film with regard to cement paste [2,29,30].

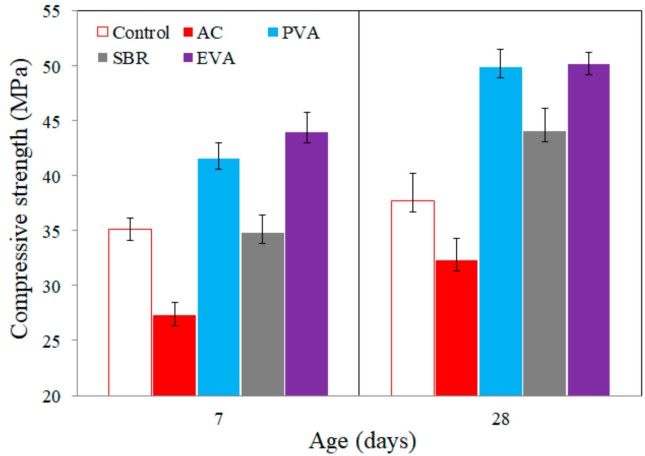

**Figure 2.** Compressive strengths for the PCMs and control mortar measured at seven and 28 days.

### 4.2. Effect of Polymer Type on Flexural Strength

Table 6 and Figure 3 illustrate the averaged flexural strengths for both PCMs and the control specimen. PCM samples ranked EVA, PVA, SBR, and AC in flexural strength from high to low. The PCM containing EVA exhibited the highest strength, and the value was 63% higher than that of the control mortar. The flexural strength of the PCM containing SBR was 21% higher than that of the PCM with AC. Medeiros et al. [28] pointed out that the effects of the polymer incorporated in the PCM can be more clearly expressed in terms of flexural strength rather than the compressive strength. The effect of the polymer investigated in this study was clear in the measured flexural strength test. Specifically, 28-day flexural strength values of the PCMs containing EVA, PVA, SBR, and AC were 63%, 46%, 41%, and 16% higher, respectively, than that of the control specimen. Poston et al. [31] recommended that the repair material should have 10% higher flexural strength compared to that of control mortar. Thus, it can be concluded that all the PCMs adopted in this study satisfy the recommendation. The increase in flexural strength between seven and 28 days varied greatly depending on the polymer type. The rates of strength increase of the PCMs ranged between 25% and 41%, while the control had only a 10% increase between seven and 28 days. In more detail, the PCMs containing EVA and SBR showed the same 41% increase in measured flexural strength. In conclusion, the PCM sample produced with EVA exhibited the best performance in flexural strength.

**Table 6.** Averaged flexural strengths measured at seven and 28 days.

| Flexural Strength (Days) | Control | AC | PVA MPa | SBR | EVA |
|---|---|---|---|---|---|
| 7 | 6.23 | 6.31 | 7.45 | 6.79 | 7.91 |
| 28 | 6.81 | 7.92 | 9.91 | 9.62 | 11.09 |

### 4.3. Relationship between Compressive and Flexural Strengths

The correlation of the two strength values was investigated and compared with previous research. Figure 4 describes the relationships between the measured compressive and flexural strengths. All averaged compressive and flexural strength values are presented on the graph. Medeiros et al. confirmed the strong correlation of the two strength values with an $R^2$ value equal to 0.90 [28]. In this

study, a strong relationship was also observed, and $R^2$ values equal to 0.77 for both seven and 28 days were observed, as illustrated in Figure 4.

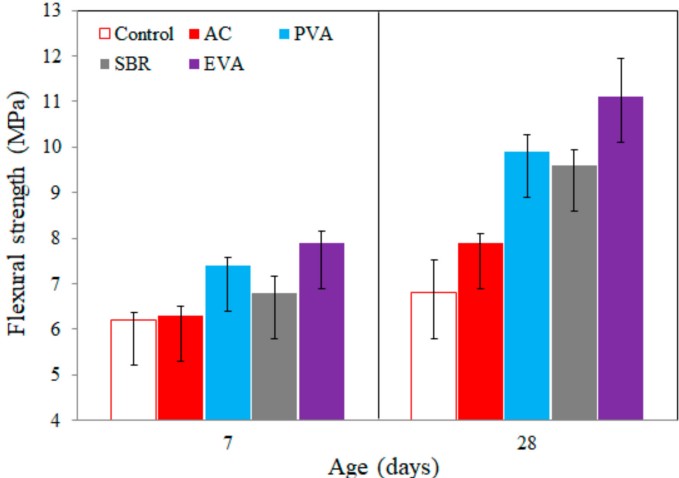

**Figure 3.** Flexural strengths for the PCMs and control mortar measured at seven and 28 days.

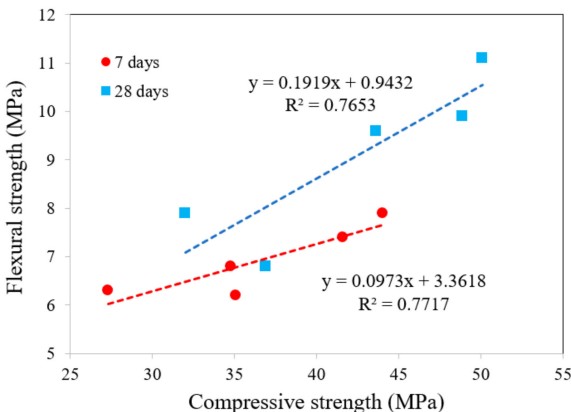

**Figure 4.** Relationships between averaged compressive and flexural strengths at seven and 28 days.

*4.4. Tensile Bond Strength and Failure Modes*

Table 7 and Figure 5 depict the effects of the polymers on the averaged pull-off bond strengths measured at seven and 28 days. PCM samples ranked SBR, PVA, EVA, and AC in tensile bond strength from high to low at 28 days. The PCM containing SBR showed the highest bond strength at 28 days, and the value was approximately 29% higher than that of the control mortar. The PCM with SBR also showed a higher bond strength than that of the PCM with AC, and this result is consistent with the previous study reported by Mirza et al. [7]. The PCM containing EVA exhibited the highest bond strength at seven days, but it had the lowest increase in the rate of bond strength between seven and 28 days. The bond strength of the PCM containing PVA was still high, and the value was 1% lower than compared to that of SBR. The failure modes of each PCM were investigated through the visual inspection of the substrate concrete specimen after finishing the bond strength measurement. Overall, two different failures of the applied PCMs or at the interface between the PCM and the substrate were predominant, and no other types of failure such as failures at the epoxy glue or substrate concrete were observed. Interestingly, the failure mode of the PCMs was transferred from the mortar to the interface failure with an increase in curing time as shown in Figure 6. This can be attributed to the increased tensile strength of the PCMs. The correlations between the measured strength values were analyzed, as illustrated in Figures 7 and 8. Figure 7 shows the relationships between the compressive and bond strength, while Figure 8 depicts the relationships between flexural and bond strength. All

averaged strength values are presented on the graphs. The measured bond strength values had a stronger relationship with flexural strength than that of compressive strength. In addition, this result is consistent with the previous research reported by Medeiros et al. [28]. In addition, stronger relationships with higher $R^2$ values were observed at seven days compared to 28 days.

**Table 7.** Averaged pull-off bond strengths measured at seven and 28 days.

| Pull-Off Bond Strength (Days) | Control | AC | PVA MPa | SBR | EVA |
|---|---|---|---|---|---|
| 7 | 2.53 | 2.47 | 3.23 | 3.23 | 3.40 |
| 28 | 2.88 | 3.41 | 3.68 | 3.71 | 3.42 |

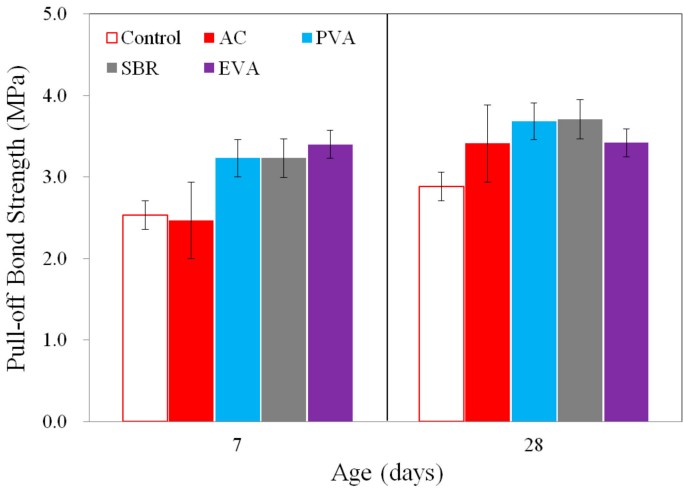

**Figure 5.** Pull-off bond strengths for the PCMs and control mortar measured at seven and 28 days.

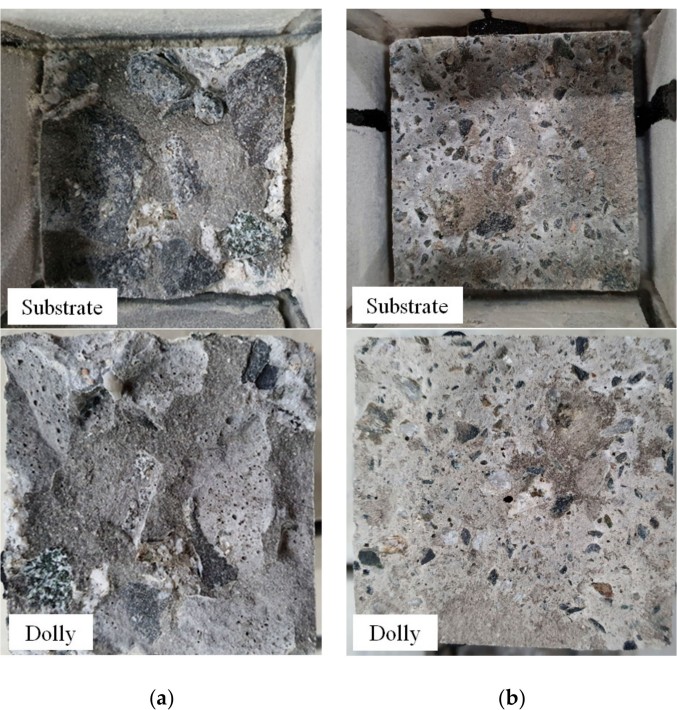

(**a**)  (**b**)

**Figure 6.** Typical failure modes: (**a**) mortar failure; (**b**) interface failure.

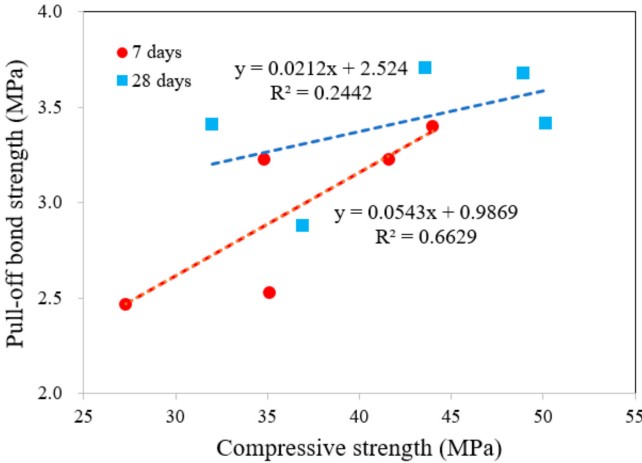

**Figure 7.** Relationships between averaged compressive and pull-off bond strengths at seven and 28 days.

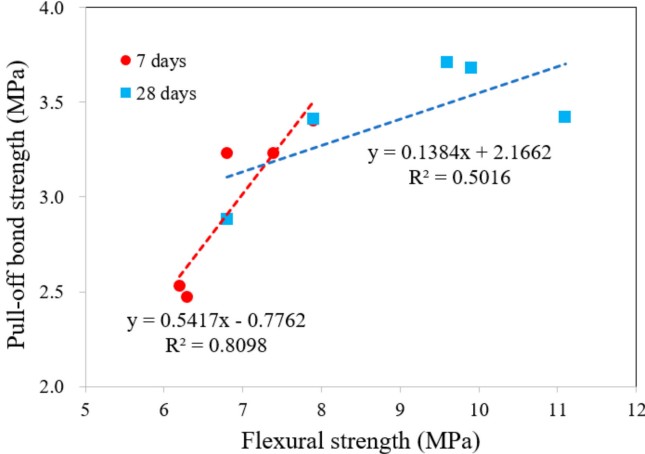

**Figure 8.** Relationships between averaged flexural and pull-off bond strengths at seven and 28 days.

## 5. Discussions and Recommendations

In this study, four different polymers (AC, PVA, SBR, and EVA) with two different dosages (0% and 3%) were employed to prepare polymer-modified cement mortars. The mechanical properties such as compressive strength, flexural strength, and pull-off bond strength were measured and compared with those of control mortars. The measured strengths were generally improved with the addition of polymers, and the results showed good agreement with previous literature. In general, the polymer-modified cement mortars containing EVA showed the best performance, with up to 33% and 63% increases in compressive and flexural strengths after 28 days, respectively, in terms of mechanical properties measured in this study. In comparison, AC, PVA, and SBR produced smaller (16%–46% compared to control) improvements in the flexural strength after 28 days. As shown in Figure 4, Figure 7, and Figure 8, the relationships between measured strength values were investigated and compared to a previous study [28]. Effects of the incorporated polymers were clearly expressed in terms of flexural strength rather than compressive strength. The reason for the better mechanical properties from the sample containing EVA is not clear at this moment, and further experimental studies must be conducted. It should be noted that the findings of this research are limited to the polymer dosage of 3% adopted in this study, and different dosages might significantly change the performance of produced PCMs. Therefore, the ranking of the investigated polymers using the limited dosage in this study should not be regarded as a general ranking for the effect of these polymers on the PCMs.

　　American Concrete Institute (ACI) committee 546 recommends the compressive strength of repair materials to be higher than 28 MPa [32]. In addition, Sprinkel reported that the tensile bond strength of repair materials should be higher than 2.1 MPa for the repair of bridge deck structures [33]. In this regard, it can be seen that the measured properties of control sample prepared in this study satisfy the guidelines and recommendations. However, it is obvious that the repair materials showing higher strength and improved interfacial bonding should be selected for the better repair work for RC structures. Furthermore, it is not appropriate to conclude the best type of polymer based on the results of this study, and further experiments must be conducted to investigate the structural performance of repaired RC members using polymeric repair materials. It is known that the mechanical properties of cement mortar can be improved with the use of polymers due to the development of the polymer film. It is also believed that the polymer film inhibits crack propagation because of its high tensile strength. The effects of polymers on the mechanical properties were confirmed throughout this study, while systematic comparisons between the different commercial polymers were not included in the scope of this study. As mentioned previously, the ranking of polymers verified in this study could not be generalized, and further experimental studies focusing on the microscopic structures and air voids of PCMs might be important.

　　Further research might be necessary for the better use of various polymers in practical applications, including patch repair, coating, floor leveling, and surface protection of concrete structures. Long-term investigations including durability and exposure tests are especially important for the selection and use of polymers. A long-term study might also be useful to determine the optimum replacement ratio of the polymer. It is known that durability performance can be improved with the use of a polymer. However, most of the previous research focused more on short-term behavior ranging between three and 28 days rather than long-term behavior. Thus, the rapid chloride penetration test (RCPT) test and carbonation depth measurement should be included in the next stage of the study. Secondly, the effects of different curing conditions on the mechanical properties of the PCMs must be studied for the possible use of polymers in marine and coastal concrete structures. Most of the previous studies, including this investigation, considered the use of conventional air-dried curing, as water immersion can influence the strength. However, marine and coastal concrete structures such as breakwaters, dock walls, and floating concrete structures are directly exposed to harsh environments, and it is expected that the concrete strength, containing polymers, might not be fully developed in such environments. Therefore, further studies must be carried out concerning curing methods to overcome the problems that can be caused due to uncontrolled conditions. Lastly, the optimum cement replacement ratio should be determined before the use of polymers in actual structures. Repeated trial tests and measurements must be used to determine the optimum cement replacement ratio and to reduce the total material cost for construction.

## 6. Conclusions

　　The tests conducted in this study aimed to clarify whether the addition of polymers affects the mechanical properties of polymer-modified cement mortars. Four commercial polymers, namely, AC, PVA, SBR, and EVA, were incorporated into a cement mortar with two different cement replacement ratios. The compressive, flexural, and pull-off bond strengths were measured, and the relationships between the measured strength values were investigated. Based on the test results and comparisons, the following conclusions can be made:

1. The present study indicated that polymer addition in a cement mortar resulted in the improvement of the compressive, flexural, and pull-off bond strength.
2. Polymers investigated in this study showed a stronger influence on the flexural strength rather than the compressive strength. Strong correlations between the compressive and flexural strength were observed.
3. Strong relationships between the flexural and pull-off bond strength were observed especially at seven days ($R^2$ = 0.81) as compared to 28 days ($R^2$ = 0.50).

4. Based on the results of the pull-off bond tests, the failure modes of the PCM samples were transferred from the mortar failure to the interface failure with an increase in curing time.
5. The PCM sample containing EVA generally showed the best performance, with up to 33% and 63% increases in 28-day compressive and flexural strengths, respectively.

The findings of this study are limited to the polymer dosage of 3% adopted, and different dosages might significantly change the performance of produced PCMs. Therefore, the ranking of the polymers investigated in this study should not be regarded as a general ranking for the effect of these polymers on the PCMs. Further studies might be necessary to determine the optimum type of polymer and cement replacement ratios for various RC structures, because the performance of the PCM can vary depending on several factors, including curing conditions, properties of the polymer, and interface preparation. In this study, all experiments and measurements were conducted under constant laboratory conditions. However, further experimental studies may consider the comparisons between the test results obtained in the laboratory and performance under real conditions.

**Author Contributions:** Conceptualization, M.O.K.; methodology, M.O.K.; validation, M.O.K.; investigation, M.O.K.; resources, M.O.K.; data curation, M.O.K.; writing—original draft preparation, M.O.K.; writing—review and editing, M.O.K.; visualization, M.O.K.; supervision, M.O.K.; project administration, M.O.K.; funding acquisition, M.O.K.

**Funding:** This research was funded by the Korea Institute of Ocean Science and Technology (KIOST), KIOST project number PE99833.

**Acknowledgments:** The experimental testing was carried out with the Department of Civil and Earth Resources Engineering at Kyoto University in Japan. The author would like to thank M. Wakasugi and T. Miyagawa. Some polymers were supplied by the Sumitomo Osaka Cement.

**Conflicts of Interest:** The authors declare no conflicts of interest.

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
