# Peer review of "Influence of Polymer Types on the Mechanical Properties of Polymer-Modified Cement Mortars"

_applsci, doi:10.3390/app10031061_

Round 1
Reviewer 1 Report
This is an experimental study showing the effect of four types of polymers on the mechanical properties of polymer modified mortars (PCM). One polymer dosage was used in the study 3%, and the effect of this dosage on the polymer-modified mortar was compared with control mortar mix with 0% polymer.
The change in the mechanical properties due to the polymeric dosage was estimated through the compressive strength test, the flexure test and the pull – off bond strength test.
In both the introduction and the research significance section, it was mentioned that the purpose of this research was to give a guide in selecting the optimum polymer type for the purpose of PCMs production. This was approached by assessing the effect of four different polymers types on the mechanical properties of the PCMs.
While the research objective is quite of interest for the civil and material engineers’ community, the methodology followed through the research restricted the findings to the polymeric dosage used in this study, hence the findings cannot be regarded as a general case in comparing these polymers effects on the produced PCMs. This is because it is well known both the type of the polymer and the dosage of the polymer, have a significant impact on the mechanical properties of the PCM.
From this, comparing the four different polymers by using one dosage is not sufficient to give a determinant conclusion and recommendation about the mechanical properties of the produced PCMs, hence not sufficient in attaining the manuscript objective, which is to recommend an optimum polymer type to be used in the repairing process unless incorporating and comparing different polymeric dosages.
The following are general comments and recommendations:
Abstract:
Line 9-Line 20:
The four polymers used in the study should be mentioned clearly in the abstract, as well as the dosage used.
Introduction:
Line 63-Line 67 :
The author referred to two studies in the literature, which both used PCMs as repairing material for cracks as well as to increase the interfacial bond with the substrate materials. It would be useful for the reader if the author included in these lines the types of the polymers used in these studies.
Experimental Setup:
Line 98
It would be useful if the physical appearance of the polymers used in the study is mentioned (latex, powder, etc.)Line 99
Type of cement should be clearly mentioned according to the relevant standard.Line 119
Normally the physical appearance of the polymer dictates the followed mixing process. This is to ensure an adequate dispersion for the polymer. The mixing process for all the polymers in this study followed the same procedure, will that affect the PCM performance results?
Test Results:
Line 158:
The enhancing effect of the added polymer on the compressive strength was attributed to ' effect of incorporated polymer'. Can this be explained more? and how some polymers such as the AC does not show this enhancement on the compressive strength?Line 203:
The failure mode was attributed for the interface failure mode. It would be useful if some photos are added to show this failure mode.Figures 3 and 5:
Including error bars is recommended to show the reliability and the repeatability of the data in the mentioned figures.Discussion:
Line 224 - Line 248
In the discussion section, it is preferably to discuss the research findings and explaining why the author thinks the different polymers types gave different results in terms of the mechanical properties. I suggest moving the recommendations made with regard to the future work to the conclusion part, and keeping the discussion section for explaining the results obtained in the research.
Reviewer 2 Report
Line 10, An experimental was Experiments were conductedLine 11, variables such as including polymer types
Line 12, what do you mean by previous results? Results in works of literature, or in your previous study?
Line 13, addition in on improving
Line 14, flexural strength rather than that on the compressive
Line 18 and 19, showed the best performance in term of mechanical properties measured in this study.
Line 19 and 20, Finally, the current status of polymers for structural applications was summarized and recommendations for future study were made.
Line 91, experimental setup Materials and methods
Line 92 and 93, An experimental setup was arranged to investigate the mechanical behavior and interfacial bond properties of PCMs.
Line 99, show the properties chemical and mineral compositions of ordinary Portland cement (OPC), the particle size distribution of fine aggregate, and properties of polymers,
Line 126 and 127, The meaning is not clear. I cannot understand this sentence.
Line 131, cubes as per ASTM C
Line 133, as per
Line 137, what do you mean by ‘for both’? Are you sure that the loading rate is 2500N/S for flexural tests? 2500N/S may be too high. Please check that.
Line 140, as per
Line 143-160, the author should describe the phenomenon that the PCM containing AC (at 7 and 28 days) and SBR (at 7 days) had lower compressive strength than control PCM.
Line 143-160, this section (section 4.1) should be written in two paragraphs. The first paragraph describes the compressive strengths. The second paragraph describes the growth rate of compressive strength from 7 to 28 days.
Line 145-146, ‘The PCM samples showed a higher compressive strength in the following order of contained polymers: EVA, PVA, SBR, and AC’ to ‘PCM samples ranked EVA, PVA, SBR, and AC in compressive strength from high to low ’
Line 158-160, this conclusion is not appropriate.
Line 166, ‘The strength was higher in the following order of polymers: EVA, PVA, SBR, and AC,’ Grammatical errors
Line 174-177, confusing
Line 223-248, this section is a literature review rather than discussions based on the results of this study. Based on the contents of Sections 4 and 5, section 5 should be deleted, and section 4 should be ‘results and discussions’.
The author should give a clear recommendation which kind of polymer should be used to modify mortar in the repair application, due to there is a short review about repaired RC structures in the ‘Section 1 Introduction, line 61-78’.
Reviewer 3 Report
The author investigated how commercial polymers affect the mechanical and interfacial properties of cement mortars. This work is important, particularly for the structural applications cited by the author. This work displays limited novelty and significance, as polymer modified cements have been investigated previously, and only a small number of substitution amounts (0 and 3%) have been tested in the present study (limited scope).
The purpose, and findings, of this work are discussed within the context of repairing reinforced concrete structures. It would be helpful to readers if the authors indicated the minimal requirements for this application. For example, how significant is the observed 63% increase in flexural strength (line 168) compare to control mortar? Is control mortar already sufficiently strong, in this loading regime, for producing or repairing reinforced concrete structures? Is this increase in strength sufficient to make the control cement suitable for repairing concrete structures? If the author includes this information, the reader would more easily understand the significance of these results.
Academic readers often quickly scan the abstract and discussion when deciding whether to read the entire paper. The authors should put numerical values in the abstract portion to increase the likelihood that readers will be engaged, and will continue to read the entire paper. The authors are of course free to express their findings as they see fit. However, I recommend revising by adding specific data, such as the following text, or something similar: “In general, the polymer-modified mortars containing ethylene vinyl acetate showed the best performance, with up to a 36% and 63% increase in compression and flexural strength after 28 days, respective, in terms of mechanical properties measured in this study. In comparison, AC, PVA and SBR produce smaller (14-46% compared to control) improvements in the flexural strength after 28 days…”
In line 140 the author should explicitly state the number of tested samples used for pull-off testing (3 samples were tested for the flexural and compression testing, as stated by the author).
In line 77, where the author describes the study, a sentence should be added justifying why they have specifically chosen 3% addition (rather than 5%, 1%, or an arbitrary amount).
The sample sizes are small (N=3), and the reader cannot interpret the significance of the results without statistical analysis indicating that the observed differences are, in fact, statistically significant. In Figure 2 the author should include the error bars for each group mean, preferably using standard deviation. A review of the cited literature (citations 1-12) confirm that error bars are not often included in similar publications. However, there are serious concerns with the present work, that the author can easily address:
a) In Figure 2, error bars must be included, otherwise comparisons cannot be made between the different polymer group means. For example, in line 148 the authors claim the EVA group is 36% stronger than control specimens. An error bar and statistical analysis would support this assertion.
b) Statistical analysis (ANOVA) should be used, if the authors wish to compare the mechanical properties of each group. For example, in line 148 the authors claim the EVA group is 36% stronger than control specimens. Without statistical analysis, the authors do not know whether the small number of samples are normally distributed and, therefore, actually representative of the true mean value for a particular group.
In line 158 the authors claim, “Overall, it can be concluded that the SBR polymer showed a more significant influence in increasing the compressive strength when compared to other polymers investigated in this study.” The authors should verify that the data presented is representative (i.e. normally distributed, etc.), which is a concern for small sample sizes. Also, when making this claim, the authors should acknowledge that each polymer will likely have a different optimal concentration, most likely related to the molar concentration rather than the weight percentage, above which the mechanical properties may be reduced. Comparing all polymers at only a single weight percentage substitution value might give the erroneous impression that one polymer produces “better” mechanical properties, when in fact slightly greater, or lesser, amounts might drastically change the performance of each polymer. Therefore, an alternative interpretation to the results in section 4.1 is that additional testing is needed, with at least 3 different non-zero concentrations of each polymer, to confirm that SBR does, in fact, produce a greater improvement in strength compared to the other polymers over a wide range of formulations. This should at least be mentioned in the results or discussion section as an alternative interpretation.
In Figure 3 and 5, the same concerns apply as in Figure 2, regarding error bars, statistics, etc.
In line 195, please revise the following sentence: “As expected, the effect of polymer addition on the improved bond strength was obvious and the strength values were higher in the following order of polymers: SBR, PVA, EVA, and AC.” Please indicate that this refers to only the 28 day data in Figure 5, and not the 7 day data. While the next sentence explains that the 28 day strength was highest for SBR, this may not be clear enough to the reader. I recommend rephrasing along the lines of ““As expected, the effect of polymer addition on the improved bond strength, after 28 days (Figure 5b), was…”
Most importantly, the main finding of this work is how 4 commercial polymers affect the mechanical properties of cement mortars. Therefore, it is most important that readers can easily understand which commercial polymers were used! In the method section the manufacturer, and specific polymer information (i.e. trade name of polymer if applicable), should be listed for each polymer. Unless the information is proprietary, the authors should provide as much information as possible on the polymers, so that their work can be cited, referenced, and built upon by others.
Reviewer 4 Report
The article in question is interesting, however several issues need to be addressed before publishing it in Applied Sciences:
Abstract
Please rewrite this part and include specific statements about your research.
3.1 Materials and Test Variables
In line 88, Author characterizes one of the variables of conducted experiment as 'cement replacement ratios' (by the polymer). However, in Table 4 (line 108), there is only one level of polymer content in designed cement mortar. This raises two questions: why was 3% chosen as the only level of polymer content in mortars by using the term 'cement replacement ratio', Author suggests that given mass of cement would be replaced with given mass of polymers. However, in Table 4, polymer-modified mortars have the same cement mass content as in the case of reference series. Please clarify. Table 4 - Concrete mix proportions - what kind of concrete is this? Please investigate the presented data - the mass amount of concrete components (water, cement, aggregate) seems to be too low for one cubic meter.4. Test results
How many specimens were prepared for each series? All figeres in this section - please include statistical information on all charts (for example - mean, lowest result and highest for each series) Line 158-160 - why SBR is better then PVA nad EVA? Both of these show higher strengths at 7 and 28 days compared to control series; why would we want to retard hydration so much? Please elaborate. Fig 4,6,7 - whiat data is presented on this graph? please correct the figure discription (include what kind of data is presented )5. Discussion
Table 5 - please delete this table and reference the information about the influence of polymers on chloride ion diffusion directly in the text.6. Conclusions
Please rewrite this part and incluede specific statement about your research. From 5 points in this section, only first one is specific enough. In remaining four, please add more specific information.Author Response
Please see the attachment.

Round 2
Reviewer 1 Report
The author should mention clearly that the findings of this paper are limited to the dosage used, as using other different dosage might change the performance the diffrent produced PCMs significantly.
Therefore the ranking of the used polymers by using the limited dosage used in this study, can not be regarded as a general ranking for the effect of these polymers on the PCMs.
Reviewer 2 Report
This manuscript is more like an experiment report rather than a scientific article that can be published. The discussion is still not enough. The mechanism of how the polymer affects the properties of mortars is still not discussed.
Reviewer 4 Report
The manuscript has been significantly improved, Authors responded to all issues raised in the previous review.
Author Response
Thank you for your time and consideration.